# Negative Correlation between Lipid Content and Antibiotic Activity in *Streptomyces*: General Rule and Exceptions

**DOI:** 10.3390/antibiotics9060280

**Published:** 2020-05-26

**Authors:** Michelle David, Clara Lejeune, Sonia Abreu, Annabelle Thibessard, Pierre Leblond, Pierre Chaminade, Marie-Joelle Virolle

**Affiliations:** 1Institute for Integrative Biology of the Cell (I2BC), CEA, CNRS, Université Paris-Saclay, 91198 Gif-sur-Yvette, France; michelle.david@i2bc.paris-saclay.fr (M.D.); clara.lejeune@i2bc.paris-saclay.fr (C.L.); 2Lipides, Systèmes Analytiques et Biologiques, Université Paris-Saclay, 92296 Châtenay-Malabry, France; sonia.abreu@universite-paris-saclay.fr (S.A.); pierre.chaminade@universite-paris-saclay.fr (P.C.); 3Université de Lorraine, INRAE, DynAMic, F-54000 Nancy, France; annabelle.thibessard@univ-lorraine.fr (A.T.); pierre.leblond@univ-lorraine.fr (P.L.)

**Keywords:** *Streptomyces*, antibiotics, lipids, triacylglycerol, metabolism, glycerol, glucose

## Abstract

Streptomycetes are well known antibiotic producers and are among the rare prokaryotes able to store carbon as lipids. Previous comparative studies of the weak antibiotic producer *Streptomyces lividans* with its *ppk* mutant and with *Streptomyces coelicolor*, which both produce antibiotics, suggested the existence of a negative correlation between total lipid content and the ability to produce antibiotics. To determine whether such a negative correlation can be generalized to other *Streptomyces* species, fifty-four strains were picked randomly and grown on modified R2YE medium, limited in phosphate, with glucose or glycerol as the main carbon source. The total lipid content and antibiotic activity against *Micrococcus luteus* were assessed for each strain. This study revealed that the ability to accumulate lipids was not evenly distributed among strains and that glycerol was more lipogenic than glucose and had a negative impact on antibiotic biosynthesis. Furthermore, a statistically significant negative Pearson correlation between lipid content and antibiotic activity could be established for most strains, but a few strains escape this general law. These exceptions are likely due to limits and biases linked to the type of test used to determine antibiotic activity, which relies exclusively on *Micrococcus luteus* sensitivity. They are characterized either by high lipid content and high antibiotic activity or by low lipid content and undetectable antibiotic activity against *Micrococcus luteus*. Lastly, the comparative genomic analysis of two strains with contrasting lipid content, and both named *Streptomyces antibioticus* (DSM 41,481 and DSM 40,868, which we found to be phylogenetically related to *Streptomyces lavenduligriseus*)*,* indicated that some genetic differences in various pathways related to the generation/consumption of acetylCoA could be responsible for such a difference.

## 1. Introduction

The *Streptomyces* genus is a well-known producer of most antibiotics in current use and of a variety of other bioactive molecules useful to human health (e.g., anti-cancer and anti-inflammatory drugs) or agriculture (e.g., fungicides, pesticides, insecticides and herbicides) [1,2,3,4]. Furthermore, *Streptomyces* is one of the rare prokaryotes to have the ability to accumulate storage lipids of the triacylglycerol family when cultivated under nitrogen and/or phosphate limitation [5]. The biosynthesis of these molecules requires the availability of glycerol3P and acetylCoA, both stemming from glycolysis. The condensation of acetylCoA results in fatty acid biosynthesis. The transfer of fatty acids on the glycerol backbone by glycerol3P acyl transferase (GPAT) yields diacylglycerol (DAG), a molecule precursor of polar membranous phospholipids as well as neutral lipids such as triacylglycerol (TAG). In previous studies, we demonstrated that, when grown on R2YE medium [6] limited in phosphate and containing glucose as the main carbon source, the weak antibiotic producer, *S. lividans*, had a high TAG content, indicative of a glycolytic metabolism [7]. By contrast, the *ppk* mutant of *S. lividans* and the closely related model strain, *S. coelicolor*—which both produce the same antibiotics—have a low TAG content compared to the wild type strain of *S. lividans* [7,8]. These two strains suffer energetic stress. *S. coelicolor* suffers energetic stress since the two component system PhoR/PhoP that positively controls the phosphate supply, necessary for ATP synthesis, is weakly expressed in this strain [9] whereas the *ppk* mutant suffers energetic stress since its lacks PPK, an important enzyme that regenerates ATP from ADP and polyphosphate [10,11]. In these two strains, similar homeostatic processes are triggered to restore their energetic balance [7,9]. Such processes include the activation of their TCA cycle, which yields reduced co-factors whose re-oxidation by the respiratory chain generates ATP. In these strains, with acetylCoA being used to feed the TCA cycle, it is not available to be stored as lipids [7,9]. The activation of the oxidative metabolism of these strains was shown to be correlated with the production of the antibiotics CDA (calcium dependent antibiotic), RED (undecylprodigiosin) and ACT (actinorhoddin) [7]. CDA and RED were proposed to create damage to the membrane, contributing to the cell death and lysis of a fraction of the population [7,12,13]. This would provide nutrients, and especially phosphate, to support the activation of the oxidative metabolism of the surviving population. Furthermore, since the onset of ACT biosynthesis was shown to coincide with an abrupt drop in the intracellular ATP concentration in *S. coelicolor*, ACT, like other molecules possessing quinone groups (melanine, humic acid etc.), was proposed to act as an electron acceptor [7]. In a context of low phosphate availability, ACT would reduce the electron flow through the respiratory chain, reducing respiration efficiency and thus ATP generation, in order to adjust it to low phosphate availability [7,12].

A putative link between TAG content and antibiotic activity was already anticipated twenty years ago by Olukoshi and Packter [14,15] in *S. lividans*. Furthermore, the impaired biosynthesis [16,17] or enhanced degradation [18,19,20] of fatty acids was indeed shown to be correlated with increased antibiotic biosynthesis in *S. coelicolor*. Similarly, enhanced spiramycin biosynthesis was correlated with lower lipid content in *S. ambofaciens* [21]. In order to determine whether these observations constituted a general rule for *Streptomyces* species, fifty-four randomly picked *Streptomyces* strains from our collection were grown on modified solid R2YE medium limited in phosphate with either glucose or glycerol as the main carbon source. Their total lipid content, on glucose or glycerol, was assessed by Fourier transformed infra-red spectroscopy (FTIRS) [22,23,24], and the measure of the diameter of the growth inhibition zones around four agar plugs of each strain gave an estimation of their antibiotic activity against *Micrococcus luteus* (Figure 1). This study confirmed the existence of a negative correlation between lipid content and antibiotic activity but with some exceptions commented on in the discussion. It also revealed that glycerol was more lipogenic than glucose but was not as good a carbon source as glucose to promote antibiotic activity. An explanation concerning this unexpected observation is also proposed in the discussion. Lastly, a comparative analysis of the genome of two *Streptomyces antibioticus* strains—DSM 41,481 and DSM 40,868, bearing among the highest and lowest lipid contents on glycerol—was carried out in an attempt to determine the genetic features responsible for their highly different lipid contents.

## 2. Results

### 2.1. The Lipid Content and Antibiotic Activity of the Streptomyces Strains Varies with the Nature of the Carbon Source Used for Growth

Fifty-four randomly picked *Streptomyces* strains from our collection were grown for 72 h at 28 °C on the surface of cellophane disks deposited on the surface of the modified (no sucrose added) R2YE medium [6] limited in phosphate and containing either glucose (50 mM) or glycerol (100 mM) as the main carbon source. The lipid content of the strains was determined using the fast and reliable FTIRS method [22,23,24]. The FTIRS values, expressed as arbitrary units, were converted into µg of Fatty Methyl Esters (FAME) per mg of dry mycelium using the converting equation established by Millan-Oropeza et al. (2017) [24]. Indeed, in the presence of methanol, fatty acids constitutive of all cellular lipids are esterified to yield FAME that can be accurately quantified by GC/MS [22,23,24]. The presence of antibiotic activity excreted in the agar medium was assessed by the measurement of the diameters of the zones of growth inhibition of *Micrococcus luteus* around four agar plugs of each *Streptomyces* strain tested. The strains were ranked according to their lipid/FAME content on glucose (Figure 1a) or glycerol (Figure 1b), and their antibiotic activity is shown in (Figure 1c). This study revealed that the average lipid content of the strains was highly variable, revealing their highly different metabolic features, and was usually higher on glycerol than on glucose (Figure 1 and Figure 2). It also revealed that some strains have far lower lipid content on glucose than on glycerol (Figure 1b). These strains might have, like *S. coelicolor*, an oxidative metabolism yielding an insufficient amount of glycerol3P to support high lipid biosynthesis on glucose [9]. Thirty and fifteen strains (forty-five strains) had a medium (50–100 µg/mg) or high (>100 µg/mg) lipid content on glycerol whereas only twenty and seven strains (twenty-seven strains) had medium or high lipid content on glucose, respectively. Consistently, the number of strains bearing a low lipid content (<50 µg/mg) was 3 fold higher on glucose (twenty-seven strains) than on glycerol (nine strains) (Figure 2). This confirmed that glycerol was more lipogenic than glucose, likely because the generation of glycerol3P, a major lipid precursor, from glycerol is more direct than that from glucose (one step versus five steps).

This study also revealed that the antibiotic activity of the strains was also highly variable and was usually higher on glucose than on glycerol (Figure 3a,c). On glucose, fifteen strains gave halos of inhibition >8 mm, and among them, nine (60%) had a low lipid content, five (33.3%) had a medium lipid content and one (6.7%) had a high lipid content. By contrast, on glycerol, only nine strains gave halos of inhibition >8 mm, and among them, only one (11.1%) had a low lipid content, six (66.7%) had a medium lipid content and two (22.2%) had a high lipid content (Figure 3a,c). Furthermore, among the thirty-four strains that gave inhibition halos with diameters greater than or equal to 4 mm, fourteen (41.2%) gave halos of similar size on both carbon sources, fourteen (41.2%) gave bigger halos on glucose than on glycerol and only six (17.6%) gave bigger halos on glycerol than on glucose (Figure 1c). This suggested that glycerol does not promote as good antibiotic activity as glucose. A hypothesis concerning the cause of this difference is proposed in the discussion.

Interestingly, on glucose, the proportion of strains yielding big halos of inhibition (>8 mm) decreases when the lipid content increases, and conversely, the proportion of strains yielding small halos of inhibition (<4 mm) increases when the lipid content increases (Figure 3b). These features, that are consistent with a negative correlation existing between lipid content and antibiotic activity, are not seen on glycerol (Figure 3d).

### 2.2. A Statistically Significant Negative Pearson Correlation between Total Lipid Content and Antibiotic Activity Could Be Established with Some Exceptions

In order to determine whether a statistically significant negative correlation between total lipid content and antibiotic activity could be established for glucose but also for glycerol, the lipid content of each strain was plotted against its respective antibiotic activity and a linear regression curve was established (Figure 4a,c). This indicated that on R2YE glucose or glycerol, no strong negative Pearson correlation (*p*-value defined as <0.05) could be established between total lipid content and antibiotic activity since *p*-values of 0.135 and 0.227 were obtained for glucose and glycerol, respectively.

However, we noticed that if strains with extreme features were removed from our statistical analysis, a negative correlation between lipid content and antibiotic activity could be established. These strains were chosen by calculating quartiles. The graph (Figure 4a), representing the fifty-four strains grown on glucose, was divided into quartiles according to their lipid content (dashed vertical lines) and antibiotic activity (dashed horizontal lines). Each quartile contains 25% of each type of data. This delineated sixteen areas in which the strains had similar features concerning their lipid content and antibiotic activity. The three strains (*S. brasiliensis*, *S. diastaticus* subspecies *diastaticus* and *S. albus*) present in the lower corner area were in the quartiles bearing the lowest lipid content and no detectable antibiotic activity. By contrast, the three strains (*S. bellus* subspecies *bellus*, *S. echinatus* and *S. aridus*) present in upper corner area were in the quartiles bearing the highest lipid content and the highest antibiotic activity. The removal of these six strains with extreme features greatly improved the significance of the negative correlation between lipid content and antibiotic activity since statistically significant *p*-values of 0.002 and 0.013 (<0.05) and Pearson correlation coefficients of −0.432 and −0.355 were then obtained for glucose- and glycerol-grown cultures, respectively (Figure 4b,d).

### 2.3. Comparison of Streptomyces Antibioticus DSM 41,481 and DSM 40,868 Strains with Drastically Different Lipid Contents

In order to get a better understanding of the molecular basis for the huge difference in lipid content observed between strains, two strains named *Streptomyces antibioticus* DSM 41,481 (GenBank Acc. NZ_CM007717.1) and DSM 40,868 (ATCC 11,891, GenBank Acc. CP050504) that have among the highest and lowest lipid content, respectively, especially on glycerol, were studied in more detail.

The growth curves on R2YE glucose indicated that the strain DSM 41,481, bearing high lipid content, reached the stationary phase earlier and had a lower biomass yield than the strain DSM 40,868 (Figure 5a). The content in all lipid classes, and not only TAG as anticipated, was, on average, 2.5 fold higher in *S. antibioticus* DSM 41,481 than in DSM 40,868 (Figure 5c), consistent with FTIRS measurements [22,23,24]. Furthermore, the strain DSM 40,868 produced an antibiotic activity against *Micrococcus luteus*, whereas DSM 41,481 did not (Figure 5b). However, the production of antimycin by the strain DSM 41,481 was detected by LC/Corona-CAD (Figure 5d and Appendix A), and the presence of an NRPS/PKS antimycin cluster in the genome of this strain was confirmed by anti-Smash [25] (Appendix A). In eukaryotes, antimycin impairs the correct functioning of the cytochrome c reductase, a key mitochondrial enzyme of the respiratory chain, and thus inhibits oxidative phosphorylation [26,27] *Streptomyces* is known to possess several cytochrome c reductases/oxidases that could possibly be antimycin targets, but no report in the literature indicates that antimycin has an impact on the respiratory activity of *Streptomyces* producing strains. However, we think that it is likely that antimycin plays such a role in the producing strain since bio-active molecules often fulfill transient but important regulatory roles in their producer [12]. Furthermore, *Streptomyces* genomes are known to encode several eukaryotic-like enzymes as well as biosynthetic pathways directing the synthesis of molecules regulating their activity as well as that of their eukaryotic orthologs [28,29,30]. Since the strain DSM41481 stops growing earlier than the other strain, we cannot totally exclude that the antimycin produced by this strain alters the correct functioning of its oxidative metabolism, resulting in a reduction of TCA cycle activity. The latter would thus consume less acetylCoA, and an excess of acetylCoA would be stored as lipids.

Since the two strains bear the same lipid classes, they most likely possess the same enzymes involved in the biosynthesis of the latter. As a consequence, the difference in their lipid content might be rather due to a difference in the availability of the necessary lipid precursor, acetylCoA. We thus examined, as a priority, through comparative genome analysis, genetic differences between the two strains in pathways involved in the generation, consumption or storage of acetylCoA that could be responsible for the high and low lipid contents of *Streptomyces antibioticus* DSM 41,481 and DSM 40,868, respectively.

### 2.4. Comparative Analysis of the Sequence of the Genome of Streptomyces Antibioticus DSM 41,481 and DSM 40,868

The genomes of the two strains of *S. antibioticus*—DSM 41,481 and DSM 40,868, called OSCP and OSBF in our collection—were fully sequenced thanks to a combination of Nanopore (Oxford Nanopore Technologies) and Illumina technologies. A hybrid assembly was generated to give a single contig per genome, of 8,473,575 bp and 9,195,693 bp for DSM 41,481 and DSM 40,868, respectively. They possess Terminal Inverted Repeats (TIR) of 26,378 bp and 13,483 bp, respectively. The genomic sequence of strain DSM 41,481 was almost identical to that present in the Genbank database (CM007717.1), with a very strong nucleotide identity over the whole genome (greater than 99.9%). Moreover, the two genomic sequences are fully collinear throughout (not shown). Due to the high quality of the sequencing performed in this work thanks to the dual technology used, we have deposited it in Genbank under the accession number CP050692.

In addition, the assignment of the strain DSM 40,868 to the species *S. antibioticus* was questioned by phylogenetic analyses carried out using the acquired genomic sequence. Indeed, its 16S rDNA sequence was identical to that of *S. lavenduligriseus* strain NBRC 13,405, while it bears only 96.47% identity with *S. antibioticus* DSM 41,481. Furthermore, a phylogenetic reconstruction showed that DSM 40,868 better clustered with *S. lavenduligriseus* strain NBRC 13,405 than with *S. antibioticus* strains. However, since DSM 40,868 is listed as *S. antibioticus* in both the DSMZ and ATCC website collections (DSM 40,868 is also registered as ATCC 11,891), re-naming this strain as *S. lavenduligriseus* would be confusing, so this strain was called *S. sp.* DSM 40,868, and its genome sequence was registered under the Genbank accession number CP050504. The annotation was performed using the RAST tool kit (RASTtk) [31], available on the Rapid Annotation using Subsystem Technology (RAST) platform. *S. antibioticus* DSM 41,481 included 7823 predicted coding DNA sequences (CDSs), while *Streptomyces* sp. DSM 40,868 contained 8637 CDSs. AntiSMASH was used to predict the presence of biosynthetic gene clusters [25]. Twenty-six and forty-six gene clusters were found in the genomes of *S. antibioticus* DSM 41,481 and *S. sp.* DSM 40,868, respectively (Appendix A).

In the comparative analysis of the two genomes, we chose to consider only the reactions catalyzed by a single enzyme that was present in one species but absent in the other.

We first noticed that the enzyme 3.1.3.10 that catalyzes the dephosphorylation of glucose 1P to glucose does not exist in the high lipid-containing strain, *S. antibioticus* DSM 41,481 (Appendix A). The absence of this enzyme might eliminate a futile ATP-consuming cycle of the dephosphorylation/re-phosphorylation of glucose and would allow a more direct and efficient conversion of Glc1P to Glc6P, favorable for the generation of acetylCoA (Appendix A).

The presence in the strain DSM 41,481, but not in DSM 40,868, of the enzyme 1.1.1.83 that catalyzes the conversion of malate into pyruvate and of 2.3.1.54 (Appendix A) that catalyzes the conversion of pyruvate into acetylCoA (Appendix A) is likely to have a positive impact on acetylCoA availability and thus on the lipid content of DSM 41,481. Furthermore, two enzymes (1.1.99.2 and 2.8.3.12) converting 2-oxoglutarate into hydroxyglutarylCoA are present in the strain DSM 41,481 but absent in DSM 40,868 (Appendix A). Since hydroxyglutarylCoA can eventually lead to the synthesis of crotomyl and butanoylCoA, precursors of odd chain or branched fatty acids, its generation might thus have a positive impact on lipid content.

Concerning nitrogen metabolism, we noted the presence in DSM 40,868 but not in DSM 41,481 of a cyanate lyase (4.2.1.104) that catalyzes the reaction of cyanate with bicarbonate, yielding carbon dioxide and NH_4_ (Appendix A). By contrast, we noted the presence in DSM 41,481 but not in DSM 40,868 of the enzyme 1.4.1.3 (Appendix A) that catalyzes the conversion of oxoglutarate into glutamate, consuming a molecule of NH_4_. Since NH_4_ is known to be inhibitory to lipid biosynthesis [32,33,34], its generation via the cyanate lyase in DSM 40,868 and its consumption, for glutamate generation, in DSM 41,481 might have negative and positive impacts on lipid accumulation in DSM 40,868 and DSM 41,481, respectively.

We also noticed that one of the multiple routes (catalyzed by the enzyme 1.1.1.169) involved in the conversion of 2-dehydropentoate into pentoate, a metabolite used for the biosynthesis of coenzyme A, is present in DSM 41,481 but absent in DSM 40,868 (Appendix A). Since coenzyme A is crucial to activate acetylCoA for fatty acid biosynthesis, the absence of this enzyme in DSM 40,868 might limit coenzyme A biosynthesis, contributing to its low lipid content.

Altogether, these genetic differences are consistent with the high and low lipid contents of *S. antibioticus* DSM 41,481 and *Streptomyces sp.* DSM 40,868, respectively. However, one cannot totally exclude that the main reason for the high lipid content of *S. antibioticus* DSM 41,481 is its production of antimycin as mentioned above.

## 3. Discussion

In this issue, we tried to determine whether a statistically significant negative Pearson correlation could be established between total lipid content and antibiotic activity in *Streptomyces* species, since previous studies suggested that such a correlation might exist [7]. The lipid content of a cell relies on its ability to synthetize glycerol3P and acetylCoA, which both stem mainly (but not exclusively) from glycolysis. AcetylCoA is a metabolic node that can have various fates. During active growth, it constitutes the main fuel of the TCA cycle, which generates the metabolic precursors of amino acids used for protein biosynthesis or other anabolic processes. However, if, for some reason (such as nitrogen or phosphate limitation), the TCA cycle and thus anabolic processes slow down, acetylCoA can then be stored as lipids, as in *S. lividans*, or used for polyketide antibiotics biosynthesis, as in *S. coelicolor*. In such a context of low anabolism, amino acids generated from intermediates of the TCA cycle can also be used for the biosynthesis of peptide antibiotics. Since the biosynthesis of these bioactive molecules directly or indirectly requires acetylCoA, their synthesis might be in direct competition with that of lipids. This could explain the negative correlation existing between these two processes, and these molecules can be seen as metabolic sinks when, in conditions of growth slowdown, the metabolites generated by anabolism exceed the needs of anabolism.

However, things might not be as simple as described above, since the biosynthesis of bioactive molecules is usually triggered by some specific physiological or environmental conditions, often linked to nutritional limitation and growth slowdown [12,35], and these molecules are believed to fulfill important regulatory roles for the producing bacteria in such contexts [12]. For instance, *S. coelicolor*, a strain characterized by an active oxidative metabolism consuming acetylCoA and thus bearing a low lipid content, produces antibiotics proposed to contribute to the regulation of its energetic metabolism [12]. Our data suggested that the existence of a negative correlation between lipid content and antibiotic activity, previously established for *S. coelicolor*, was also true for most *Streptomyces* strains, at least on R2YE glucose. However, the existence of exceptions to this general rule was noted. These strains either bear a low lipid content and undetectable antibiotic activity (Type I: *S. brasiliensis*, *S. diastaticus* subspecies *diastaticus* and *S. albus*) or a high lipid content and high antibiotic activity (Type II: *S. bellus* subspecies *bellus*, *S. echinatus* and *S. aridus*). One should be aware that assessing antibiotic activity via the measurement of the diameter of the growth inhibition zones of *Micrococcus luteus* might not reveal all antibiotic production. Indeed, the existence of Type I strains could be easily explained by the insensitivity of *Micrococcus luteus* to the antibiotic produced, which is thus not detected. This was exemplified by the production of antimycin by DSM41481, which has no impact on *Micrococcus* growth. By contrast, Type II strains may produce very small amounts of a specific antibiotic that *Micrococcus luteus* is highly sensitive to or produce low levels of multiple antibiotics that synergistically impact *Micrococcus luteus* growth etc. Anyhow, if antibiotic biosynthesis is triggered in a strain with a low to medium lipid content, antibiotic biosynthesis is likely to have a significant impact on its lipid content, but such an impact will be hardly visible in a strain bearing high lipid content.

Furthermore, to explain the unexpected lower antibiotic activity on glycerol than on glucose, we propose that oxidative stress might be less severe on glycerol than on glucose since the catabolism of glycerol—and more precisely its conversion into dihydroxyacetone by a glycerol dehydrogenase or into glyceraldehyde by a glyceraldehyde reductase—generates more NADPH than the catabolism of glucose [36,37]. NADPH is an indispensable reduced co-factor necessary to combat and thus reduce oxidative stress [38], since some reports in the literature mention that the biosynthesis of specific antibiotics might be triggered by oxidative stress [39,40,41,42,43]. The biosynthesis of such antibiotics is thus expected to be reduced on glycerol. Antibiotics of this class (defined as class II in Virolle, 2020 [12]) are thought to have anti-oxidant functions via their ability to capture electrons from the respiratory chain to prevent the formation of ROS or NOS, but in doing so, they would also be inhibitory to respiration and thus toxic for living cells.

Lastly, the comparative genomic analysis of two strains—both named *Streptomyces antibioticus*, DSM 41,481 and DSM 40,868—that we found to be more phylogenetically related to *Streptomyces lavenduligriseus* indicated that some genetic differences in various pathways related to the generation/consumption of acetylCoA could be responsible for such differences.

## 4. Materials and Methods

### 4.1. Bacterial Strains, Media and Growth Conditions

Most of the fifty-four strains used in this study were ATCC/NRLL/DSM strains. The few that do not come from these collections (*S. albus*, *S. spectabilis*, *S. actuosus*, *S. coelicolor* M145, *S. lividans* TK24, *S. pristinaespiralis* and *S. rimosus* 2535) were generous gifts from John Innes Institute. Viable spores (10^6^) of each strain were plated on the surface of four petri dishes (42 mm diameter) of solid modified R2YE medium [44] covered by a cellophane disk (Focus Packaging & Design Ltd., Louth, UK). No sucrose and no K_2_HPO_4_ were added to this modified solid R2YE medium supplemented with either glucose (50 mM) or glycerol (100 mM) as the main carbon sources. However, the determination of the content of free phosphate in this medium with a PiBlue phosphate assay kit (Gentaur, France) revealed a concentration of free Pi of 1 mM, corresponding to a situation of phosphate limitation. The plates were incubated for 72 h at 28  °C. After 72 h of incubation, mycelia were scraped off the cellophane disks of each plate with a spatula and lyophilized in order to assess their total lipid contents by Fourier transform infra- red spectroscopy (FTIRS) [22,23,24]. Four agar plugs of each plate were deposited on a lawn of the Actinobacteria *Micrococcus luteus* in order to determine the antibiotic activity of the *Streptomyces* strains against this strain that is known to be sensitive to most antibiotics [45,46]. The growth curves of *Streptomyces antibioticus* DSM 41,481 and *Streptomyces sp.* DSM 40,868, the two strains with extremely high and low total lipid contents, were obtained on the same R2YE medium (Figure 5a), and their lipid content was assessed by LC/Corona-CAD (Figure 5d).

### 4.2. Determination of Total Lipid Content Using Attenuated Total Reflectance-Fourier Transform Infra Red Spectroscopy (ATR-FTIRS) Measurements

In order to determine total lipid content, lyophilized mycelial samples of the *Streptomyces* strains were subjected to FTIR spectroscopy using a Bruker Vertex 70 FTIR spectrophotometer with a diamond ATR attachment (PIKE MIRacle crystal plate diamond ZnSe) and an MCT detector with a liquid nitrogen cooling system. A reference spectrum resulting from one hundred averaged scans obtained in absence of any sample on the Infra Red support was acquired before each sample analysis. Scans were conducted from 3600 cm^−1^ to 600 cm^−1^ with a spectral resolution of 4 cm^−1^, with 100 averaged scans for each sample. This technique allows the establishment of spectral fingerprints of the complex biological structures under investigation. The C-H stretching bands of the CH_2_ of fatty acid chains (between 2959 and 2852 cm^−1^) and the C=O ester stretching band of the ester carbonyl (band near 1740 cm^−1^) are characteristic of lipids including mainly polar membrane lipids and neutral lipids such as TAG. The height of the sharp and distinct C=O ester stretching band of the ester carbonyl is especially relevant for monitoring the total intracellular lipid contents of the strains [23]. Furthermore, since the biomass protein content can be directly characterized by the amplitude of the Amide I absorption band (1650 cm^−1^), all the FTIR spectra can be normalized to this band, allowing the comparison of the total lipid contents of the mycelial lawns of different strains. The total lipid contents of the strains assessed by FTIRS, expressed as arbitrary units, was converted into µg of Fatty Methyl Esters (FAME) per mg of dry mycelium using the converting equation established by Millan-Oropeza et al. (2017) [24]. Indeed, in the presence of methanol, fatty acids constitutive of all cellular lipids are esterified to yield FAME that can be accurately quantified by GC/MS [22,23,24]. The differences in the lipid contents of the strains were clear and highly reproducible.

### 4.3. Determination of the Antibiotic Activity of the Strains against Micrococcus Luteus

In order to determine the inhibitory impact that each *Streptomyces* strain had on *Micrococcus luteus* growth, four agar cylinders were taken aseptically from each of the replica plates using an appropriate device and were deposited onto the surface of LB agar plates, prepared as follows. Twenty-five microliters of a fresh culture of *Micrococcus luteus* (OD 600 nm of 0.4) were added to 3 mL of soft nutrient agar (SNA) that was poured onto the surface of the LB agar plate. As soon as the SNA containing *Micrococcus luteus* was solidified, the agar plugs were deposited on the surface of the plates and incubated 24 h at 37 °C. For each plate, the size of the four zones of inhibition was measured and reported in an Excel file, and the mean was calculated.

### 4.4. Statistical Analysis of the Correlation between the Lipid Content and Antibiotic Activity of Studied Streptomyces Strains

The total lipid contents of the strains were expressed as mean ± standard deviation in bar graphs and were subjected to the ANOVA test (Figure 1a,b). Correlation analyses between lipid content and antibiotic activity were achieved using Pearson correlation (Figure 4). All statistical methods and data representation were conducted in R 3.3.2 [47], using the “RVAideMemoire” package [48]; a *p*-value < 0.05 was considered as statistically significant.

### 4.5. Lipid Extraction and Characterization by LC/Corona-CAD and LC/MS

Lipid extraction was performed by a procedure derived from Folch’s method [49] from four independent cultures of *Streptomyces antibioticus* DSM 41,481 and DSM 40,868. A defined volume (4.5 mL) of chloroform/methanol (1:2) was added to 10 mg of lyophilized *Streptomyces* mycelium and vortexed for 30 s. The mixture was left at ambient temperature for 1 hour, then 1.25 mL of water was added, and the mixture was vortexed for 30 s. The mixture was then centrifuged (1000× *g* for 10 min) to obtain phase separation. The lower organic phase was collected, and the upper aqueous phase was submitted to a second extraction by adding 2 mL of chloroform/methanol (85:15). The two organic phases were pooled and evaporated under a stream of nitrogen at room temperature. The dry residue was dissolved in 400 µL of isooctane/chloroform (4:1) before analysis. The chromatographic conditions have been described previously [50]. Briefly, lipid class analysis was performed with an Inertsil Silica (150 mm × 2.1 mm I.D, 5 µm) column (GL Sciences Inc., Tokyo, Japan) thermostated at 40 °C. The HPLC instrumentation consisted of the system Dionex U-3000 RSLC (Thermofisher, Villebon, France). A quaternary solvent gradient (Appendix A) was used to elute all the lipid classes present in the sample by increasing the order of polarity. Lipid class identification was verified by coupling the chromatographic separation to mass spectrometry. MS analyses were performed with a LTQ-Orbitrap Velos Pro (Thermo Fisher Scientific) equipped with an APPI ion source. The MS^2^ and MS^3^ spectra were obtained in data-dependent acquisition (DDA) mode. Lipid detection was performed using a Corona-CAD system (ESA, Chelmsford, MA, USA) [50]; the signal was acquired with a Chromeleon data station (Thermo Fisher Scientific, Villebon-sur-Yvette, France). Corona-CAD is a universal detector used for liquid chromatography and described in Dixon and Peterson [51]. The differences in the composition of the lipid classes in the samples are expressed as peak areas. All the data were subjected to Student’s t-test using R 3.3.2 [47] and the “multcompView” package [52]. The results obtained are presented as the mean ± standard error; a *p*-value < 0.05 was considered as statistically significant.

### 4.6. DNA Isolation, Genome Sequencing and Annotation

After growth in liquid Hickey-Tresner medium [6] at 30 °C for 30 h, DNA purification was performed using the salting-out method [6], followed by chloroform extraction. A hybrid assembly using Oxford Nanopore technology for scaffolding and Illumina technology for sequence improvement was performed. Nanopore reads were generated on a gridION system. The coverage was 30x and 67x for strains DSM 41,481 and DSM 40,868, respectively. The Illumina paired-end libraries were sequenced using a MiSeq system (Illumina). The coverages of the paired-end reads (length, 301 bp) ranged from 249× to 312×. The hybrid assembly was performed using Unicycler [53]. The accession numbers are CP050692 and CP050504 for *S. antibioticus* DSM 41,481 and *Streptomyces* sp. DSM 40,868, respectively. Sequencing and assembly were performed via the I2BC NGS platform (France). The automatic annotation of the genome sequences was achieved using the RAST tool kit (RASTtk [31]) available on the Rapid Annotation using Subsystem Technology (RAST) platform. Furthermore, the annotation files were used as the input for antiSMASH [25] in order to predict the biosynthetic gene clusters content of each genome. Biosynthetic pathways were mapped with KEGG, using the GHOSTZ search program and the BBH (bi-directional best hit) assignment method [54]. We used the default gene data set, to which we added the data sets available for five *Streptomyces* species (*Streptomyces coelicolor*, *Streptomyces avermitilis*, *Streptomyces griseus*, *Streptomyces scabiei* and *Streptomyces noursei*). The presence/absence of specific enzymatic steps was manually checked using BLAST searches [55].

## Figures and Tables

**Figure 1 antibiotics-09-00280-f001:**
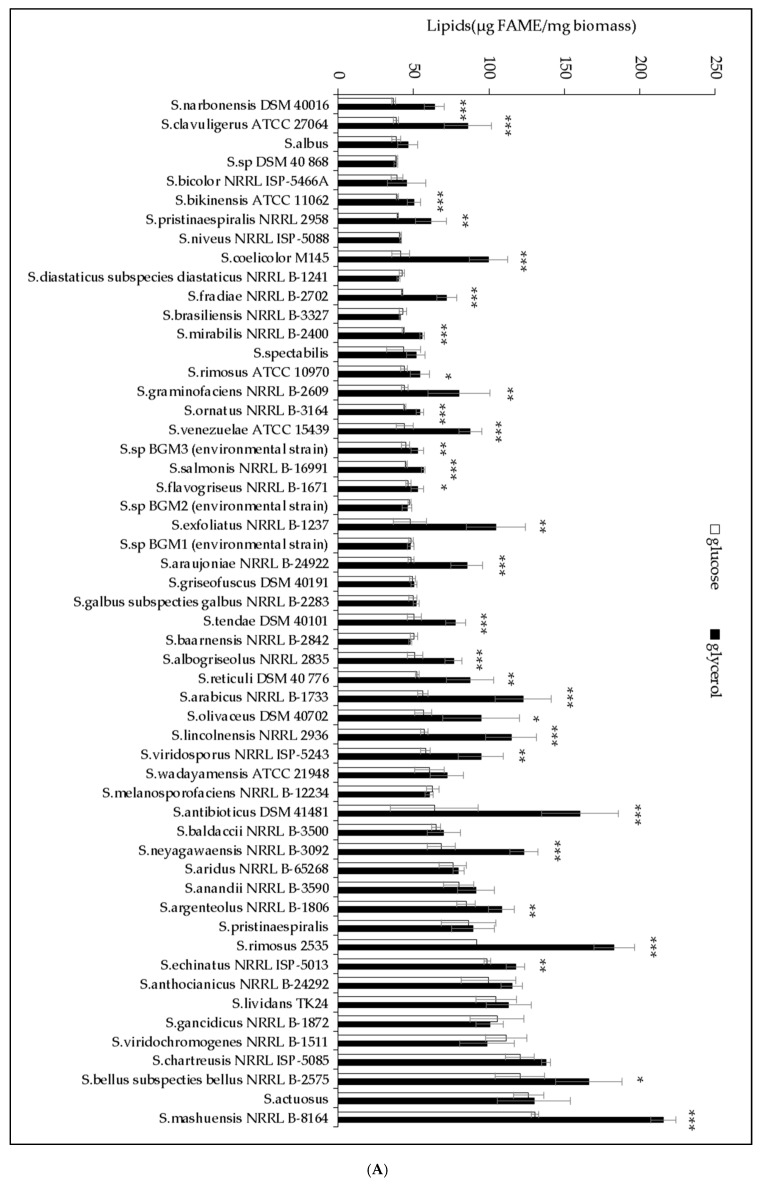
Total lipid/fatty methyl ester (FAME) content and antibiotic activity of *Streptomyces* strains. The strains were grown on modified solid R2YE limited in phosphate for 72 h at 28 °C with either glucose (white histograms) or glycerol (black histograms) as the main carbon source. Their total lipid/FAME content, determined with FTIRS, was ranked according to their increasing lipid/FAME content upon growth on (**A**) glucose or on (**B**) glycerol. Significant differences in lipid content between glucose- and glycerol-grown cultures are represented by an asterisk (ANOVA, *** = *p* < 0.001; ** = *p* < 0.01; * = *p* < 0.05). (**C**) Antibiotic activity of *Streptomyces* strains determined by the measurement of the diameters of the zones of growth inhibition of *Micrococcus luteus* around four agar plugs of each *Streptomyces* culture. Fold changes in antibiotic activity (>1.2 fold) between glucose and glycerol for inhibition zones > 4mm are shown above the histograms. The fold changes of the six cases where antibiotic activity was higher on glycerol than on glucose are boxed.

**Figure 2 antibiotics-09-00280-f002:**
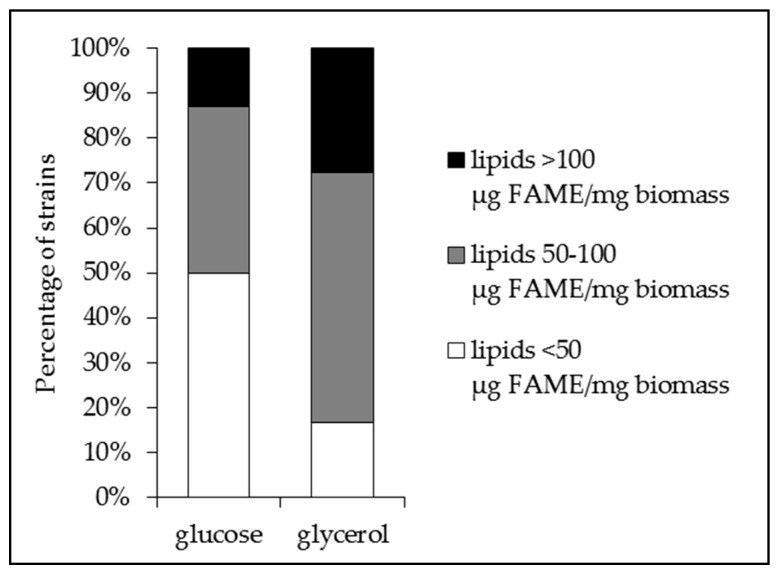
Percentage of *Streptomyces* strains grown for 72 h at 28 °C on solid modified R2YE limited in phosphate with either glucose (50 mM) or glycerol (100 mM) as the main carbon source bearing low (<50 µg FAME/mg biomass, white part); medium (50–100 µg FAME/mg biomass, grey part) and high (>100 µg FAME/mg biomass, black part) total lipid/FAME content.

**Figure 3 antibiotics-09-00280-f003:**
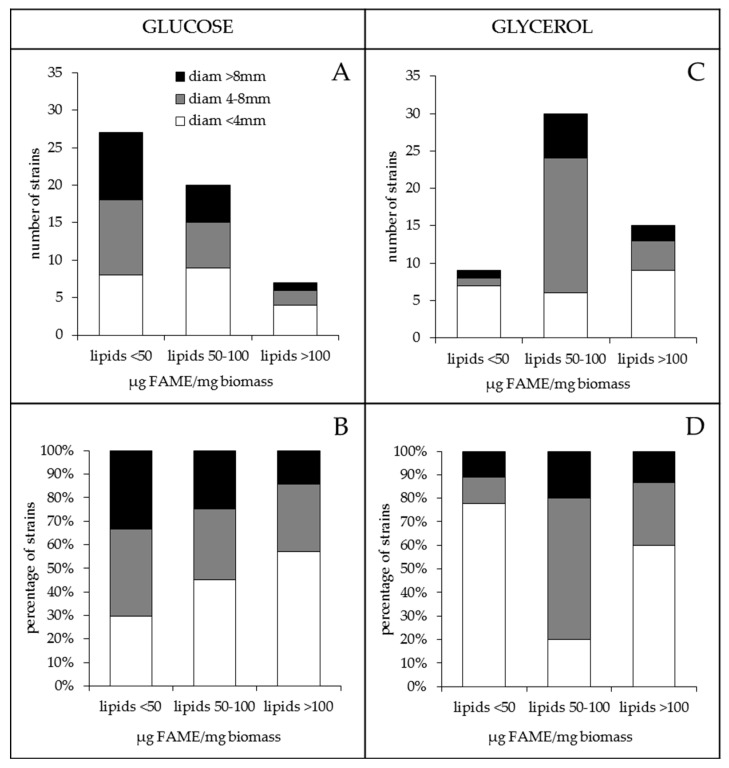
Classification of *Streptomyces* strains grown for 72 h at 28 °C on solid modified R2YE limited in phosphate, according to low (<50 µg FAME/mg biomass), medium (50–100 µg FAME/mg biomass) or high (>100 µg FAME/mg biomass) total lipid/FAME content, on glucose (**A**,**B**) and glycerol (**C**,**D**), respectively. Each lipid class is represented by an histogram, and the numbers (**A**,**B**) or percentages (**B**,**D**) of strains with weak (<4 mm), medium (4–8 mm) and strong (> 8 mm) antibiotic activity against *Micrococcus luteus* correspond to the white, grey and black parts of each histogram, respectively, on glucose (**A**,**B**) and glycerol (**C**,**D**).

**Figure 4 antibiotics-09-00280-f004:**
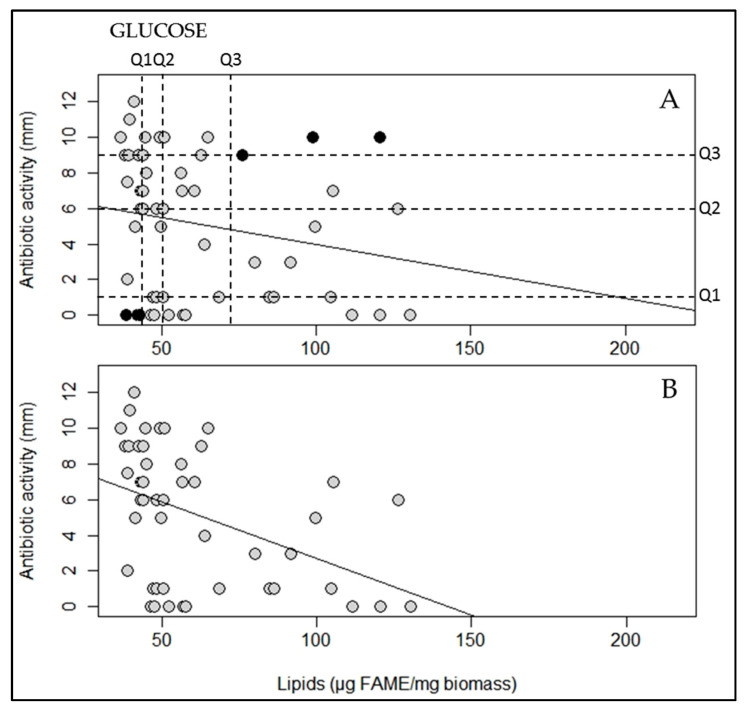
Lipid/FAME content of fifty-four *Streptomyces* strains grown on modified solid R2YE limited in phosphate for 72 h at 28 °C with (**A**) 50 mM glucose or (**C**) 100 mM glycerol as the main carbon source was plotted against their antibiotic activity, and linear regression lines were calculated (black lines). In (**A**,**C**), FAME content and antibiotic activity did not show a statistically significant negative Pearson correlation (coefficients of −0.206 and −0.167, *p*-values of 0.135 and 0.227 (> 0.05), on glucose and glycerol, respectively). In an attempt to enhance the amplitude of this negative correlation, six strains constituting exceptions (black spots) were removed. These strains were chosen as described in the text, and the vertical and horizontal dashed lines represent the quartiles (Q1, Q2 and Q3) of lipid content and antibiotic activity, respectively. The lipid/FAME content of the forty-eight remaining strains grown in the presence of (**B**) 50 mM glucose or (**D**) 100 mM glycerol was plotted against their antibiotic activity, and linear regression lines were calculated (black lines). In (**B**,**D**), the FAME content and the antibiotic activity show a statistically significant negative Pearson correlation (coefficients of −0.432 and −0.355, *p*-values of 0.002 and 0.013 (< 0.05), on glucose and glycerol, respectively).

**Figure 5 antibiotics-09-00280-f005:**
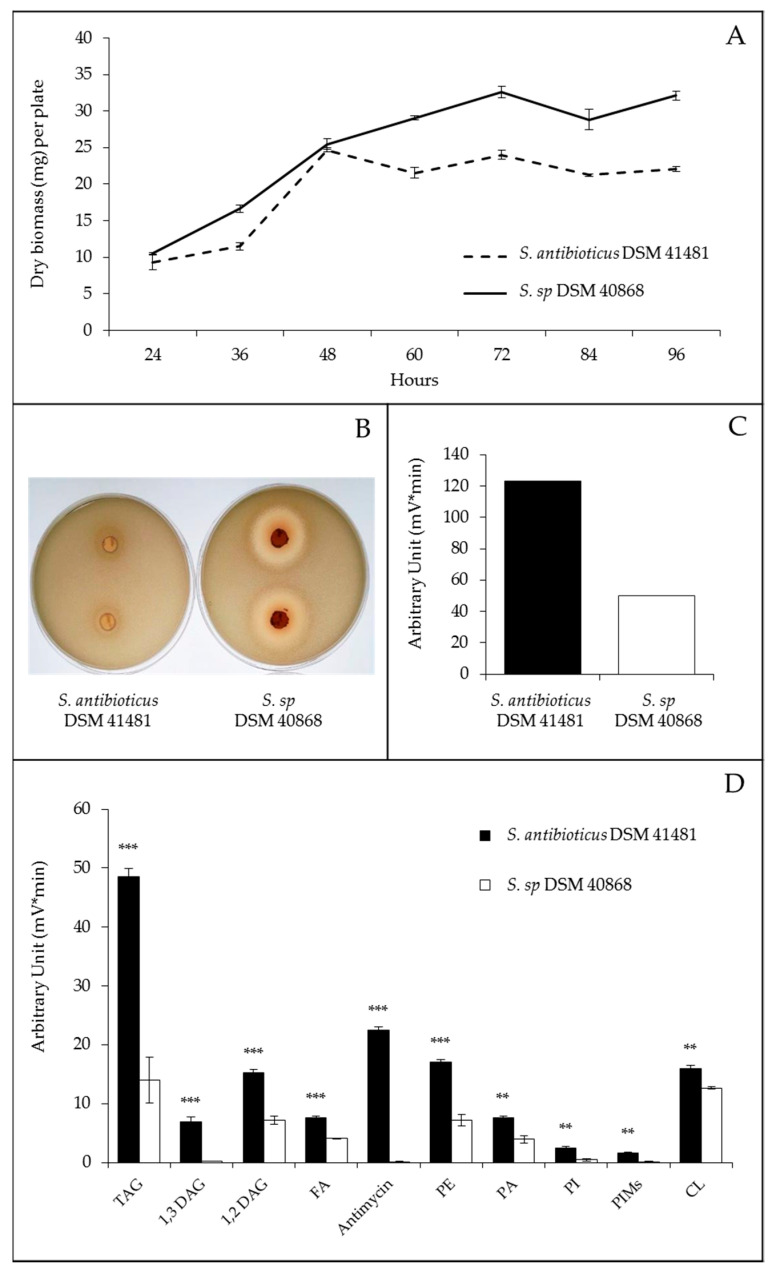
Cultures of *S. antibioticus* DSM 41,481 and *S. sp.* DSM 40,868 grown at 28 °C on solid R2YE limited in phosphate with glucose (50 mM) as the main carbon source. (**A**) Growth curves of *S. antibioticus* DSM 41,481 (dotted lines) and *S. sp.* DSM 40,868 (continuous lines); (**B**) Pictures of the zones of inhibition of *Micrococcus luteus* growth around agar plugs of 72 h-grown cultures of the *Streptomyces* strains; (**C**) Sum of all lipid classes (antimycin not included) in *S. antibioticus* DSM 41,481 (black histograms) and *S. sp.* DSM 40,868 (white histograms) according to LC/Corona-CAD data analysis; (**D**) Analysis of the total lipid contents in *S. antibioticus* DSM 41,481 (black histograms) and *Streptomyces sp.* DSM 40,868 (white histograms) grown for 72 h on solid R2YE glucose medium limited in phosphate according to LC/Corona-CAD. TAG stands for triacylglycerol; DAG, for diacylglycerol, FA for fatty acids; PE, for phosphatidylethanolamine; PA, for phosphatidic acid; PI, for phosphatidylinositol; PIMs, for phosphatidylinositol mannosides; and CL, for cardiolipid. Antimycin was detected in *S. antibioticus* DSM 41,481 but not in *Streptomyces sp.* DSM 40,868. Significant differences in lipid content between the two strains are represented by asterisks (ANOVA, *** = *p* < 0.001; ** = *p* < 0.01).

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
