# Peer review of "Negative Correlation between Lipid Content and Antibiotic Activity in Streptomyces: General Rule and Exceptions"

_antibiotics, 2020, doi:10.3390/antibiotics9060280_

Round 1

Reviewer 1 Report

The topic is interesting and the manuscript is generally written well. I found the following minor English errors:

Line 21: heavenly > heavily

Line 28: At last > Lastly

Line 166: delete "two"

Line 282: bio-actives > bioactive

Line 286: In any cases > In any case

The main problem of this work is the conclusion that the authors draw, as stated in the abstract and the discussion section:

"In conclusion, our data indicated the existence of a negative correlation between lipid content and antibiotic production..."

Authors determined antibiotic activity, and not antibiotic production. They determined antibiotic activity by measuring the zone of inhibition using Micrococcus luteus. This method is fine but the resulting data does not directly show the difference in the ability of the 54 streptomyces strains to produce antibiotics. For example, some strains may be producing a large quantity of an antibiotic that Micrococcus luteus is mostly insensitive to. Alternatively, some strains may be producing a very small amount of an antibiotic that Micrococcus luteus is highly sensitive to. Also, some strains may be producing multiple antibiotics while others are producing only one. These potential variations must be somehow accounted for.

Streptomycetes produce different types of antibiotics - polyketide, peptide, polysaccharide, etc. Does the carbon source and the overall lipid content affect all these antibiotic types? The data must be further deconvoluted.

Reviewer 2 Report

I think this work has been well performed and it has good interest in the Streptomyces field.

However, I have some questions and small comments for the authors.

1. Could you explain a bit more about the origin of the strains of your collection?
2. In line 46-47,99 and most of the times that this is mentioned, the R2YE medium modified is explained as (1mM, no K2HPO4 added), I think is more clear to write (1 mM K2HPO4 no added) unless I misunderstood what do you want to explain.
3. In line 77 there is an extra parenthesis
4. In line 99, in my opinion is more clear if it is included already what is black and white histograms, after first mention of glucose and glycerol concentrations.
5. In figures, after the main description. As the descriptions are quite long, I think is easier for the reader that the letters A, B, C, D, were before the description not after.
6. For me it is not clear the difference between Fig 1. A and B.
7. In figure 3, it would be also easier to understand in a first glance if you insert inside the plot that A-B are glucose, and B-C are glycerol (then the reader has not to go to the description that much).
8. Line 246 there is a coma that it should be after respectively and not before.
9. Line 351 it should have a coma after growth.
10. I suggest to add bioassays with more sensitive strains (not only Micrococcus luteus) in order to make stronger this correlation and test better the exceptions.
11. Do you think that this correlation takes place only in lower nitrogen or phosphate conditions? Or it is just that the correlation is more strong under this limitations?

Reviewer 3 Report

In this manuscript, David et al. test the hypothesis that intracellular lipid concentration is inversely related to antibacterial bioactivity. The authors quantified lipid content for ~50 strains using two growth media (one glucose-based, the other glycerol-based) and concluded that the glycerol-based medium was generally more lipogenic. The authors also assessed bioactivity against M. luteus using agar plugs obtained from the same culture media. The authors surmise that on the whole strains that accumulated more lipid were less bioactive. The authors sequenced the genomes of a ‘high’ and a ‘low’ lipid producer and sought to identify the genetic basis for the difference in lipid accumulation. Until now, I have not come across the ‘lipid hypothesis’ and so therefore read the manuscript with interest.

The authors have made a valiant effort to answer their question, but for me, the data lacks the robustness required to confidently conclude that lipid production/accumulation is inversely related to antibacterial bioactivity (and by extension, their production). I can see that there is a statistically significant negative Pearson correlation, but this is very weak at best, and was only achieved when removing ‘outliers’. Removal of outliers in and of itself is not a problem, however it is not clear how these outliers were selected. As an example of this, lines 159-160 indicate that S. brasiliensis and S. diastaticus subsp. diastaticus were removed because they possess low lipid content and exhibit zero antibiotic activity. These strains are represented by two black circles in Figure 4A (lower left corner) -- there are clearly three other strains adjacent to these circles that ‘fit’ the removal criteria as well.

I appreciate that not all trends in biology are quantifiable and not all correlations are 1:1, so there may be something here, but even the authors themselves point out in the discussion that interpretation of antibacterial activity is muddled when considering the physiological context of the experiment. For instance, some of the strains tested are almost assuredly producing peptide antibiotics (NRPS derived or lanthipeptides, for instance). This fact is very likely influencing the poor correlation observed for lipid content and antibacterial activity. In other words, the authors are looking at ‘total activity’, but are only quantifying lipid accumulation, which is predominantly related to only two biosynthetic systems (terpenes and polyketides). Hindsight is 20/20 and these experiments would have also been my starting point, but for me I do not feel the work adequately addresses the initial question and I therefore cannot support publication of this manuscript at present.

Other comments:

Concerning the comparison of S. antibioticus strains, a less than convincing argument is given for antimycin causing a ‘re-wiring’ of metabolism due to impairment of oxidative phosphorylation. It was shown in the 1970s that antimycin is not toxic to bacteria (B. subtilis in this case; Kei Arima & Eiji Sato (1970) Mechanism of Antimycin A Resistance in Bacillus subtilis, Agricultural and Biological Chemistry, 34:5, 739-746).   Moreover, 100 Streptomyces strains produce antimycin and its production, at least in S. albus species, is co-regulated with production of a very large polyene polyletide compound (i.e. a compound whose production would draw considerable resource from the acyl-CoA pool). The above makes it difficult, for me at least, to go along with your suggestion.

Minor things:

The acronym ‘FAME’ needs to be defined early-on in the manuscript.

Line 15:  It should be ‘which’ not ‘that’ here.

Line 20: “...activity against Micrococcus luteus for each strain was assessed.”  and remove and “plotted on the same graph.”

Line 39: prokaryotes

Line 40: instead of ‘in condition of’ think about saying, “when cultivated under...”

Line 49: most readers will not know what the ppk mutant so this should be defined briefly.

Line 50: Space required in S.lividans

Line 64: Strains from our collection (not of our collection).

Line 114-116:  I found this sentence rather difficult to get through, consider modifying it for readability/flow.

Round 2

Reviewer 1 Report

I feel that the revised conclusions are acceptable for publication. 

Author Response

The suggested changes were incorporated in the revised manuscript.

Reviewer 3 Report

Reviewer's comments/questions addressed..

Author Response

(The authors gave the same response as above.)
